# Decreased Consumption of Added Fructose Reduces Waist Circumference and Blood Glucose Concentration in Patients with Overweight and Obesity. The DISFRUTE Study: A Randomised Trial in Primary Care

**DOI:** 10.3390/nu12041149

**Published:** 2020-04-19

**Authors:** Santiago Domínguez-Coello, Lourdes Carrillo-Fernández, Jesús Gobierno-Hernández, Manuel Méndez-Abad, Carlos Borges-Álamo, José Antonio García-Dopico, Armando Aguirre-Jaime, Antonio Cabrera-de León

**Affiliations:** 1Centro de Salud La Victoria de Acentejo-Tenerife, Calle Domingo Salazar 21, 38380 Tenerife, Canary Islands, Spain; 2Unidad de Investigación de la Gerencia de Atención Primaria de Tenerife y del Hospital Universitario Nuestra Señora de la Candelaria, Santa Cruz de Tenerife, 38310 Tenerife, Canary Islands, Spain; 3Centro de Salud La Victoria de Acentejo, 38380 Tenerife, Canary Islands, Spain; lourdescarrillofernandez@gmail.com; 4Centro de Salud de Güimar 38500Tenerife, Canary Islands, Spain; jgobierno@gmail.com; 5Centro de Salud de la Orotava San Antonio, 38313 Tenerife, Canary Islands, Spain; mmendezabad@gmail.com; 6Centro de Salud Ruiz de Padrón, 38002 Tenerife, Canary Islands, Spain; cborgesalamo@hotmail.com; 7Laboratorio del Complejo Hospitalario Universitario de Canarias, 38320 La Laguna, Canary Islands, Spain; jgardop@gobiernodecanarias.org; 8Health Care Research Institute of the Santa Cruz de Tenerife College of Nursing, Colegio Oficial de Enfermeros de Santa Cruz de Tenerife, 38001 Tenerife, Spain; armagujai@gmail.com; 9European University of the Canary Islands–Member of Laureate International Universities, 38001 Tenerife, Canary Islands, Spain; 10Unidad de Investigación de la Gerencia de Atención Primaria de Tenerife y del Hospital Universitario Nuestra Señora de la Candelaria, Santa Cruz de Tenerife, 38010 Tenerife, Canary Islands, Spain; acableon@gmail.com; 11Departamento de Medicina Preventiva y Salud Pública de la Universidad de la Laguna, 38200 La Laguna, Canary Island, Spain

**Keywords:** fructose, insulin resistance, sugars, clinical trial, obesity, non-alcoholic fatty liver disease (NAFLD), primary health care

## Abstract

The relationship between fructose intake and insulin resistance remains controversial. Our purpose was to determine whether a reduction in dietary fructose is effective in decreasing insulin resistance (HOMA2-IR). This field trial was conducted on 438 adults with overweight and obese status, without diabetes. A total of 121 patients in a low fructose diet (LFD) group and 118 in a standard diet (SD) group completed the 24-week study. Both diets were prescribed with 30–40% of energy intake restriction. There were no between-group differences in HOMA2-IR. However, larger decreases were seen in the LFD group in waist circumference (−7.0 vs. −4.8 = −2.2 cms, 95% CI: −3.7, −0.7) and fasting blood glucose −0.25 vs. −0.11 = −0.14 mmol/L, 95% CI: −0.028, −0.02). The percentage of reduction in calorie intake was similar. Only were differences observed in the % energy intake for some nutrients: total fructose (−2 vs. −0.6 = −1.4, 95% CI: −2.6, −0.3), MUFA (−1.7 vs. −0.4 = −1.3, 95% CI: −2.4, −0.2), protein (5.1 vs. 3.6 = 1.4, 95% CI: 0.1, 2.7). The decrease in fructose consumption originated mainly from the reduction in added fructose (−2.8 vs. −1.9 = −0.9, 95% CI: −1.6, −0.03). These results were corroborated after multivariate adjustments. The low fructose diet did not reduce insulin resistance. However, it reduced waist circumference and fasting blood glucose concentration, which suggests a decrease in hepatic insulin resistance.

## 1. Introduction

Mortality and potential years of life lost due to cardiovascular disease and diabetes have increased in recent decades [1]. Some studies have established a direct relationship between insulin resistance (IR) and both of these diseases [2,3]. Moreover, a meta-analysis of studies involving participants without diabetes concluded that fructose intake under isocaloric conditions or with hypercaloric supplementation favored the development of hepatic IR in adults, without affecting muscular or peripheral insulin sensitivity [4]. Most studies have compared fructose versus glucose or starch matched for energy intake, or fructose versus diet alone for hypercaloric comparisons. In general, the number of participants involved in these studies was small, and when fructose was investigated, it was consumed in liquid form. Some studies were done in people with normal bodyweight, whereas others involved people with overweight or obesity [4]. A further consideration is that currently, the relationship between sugar intake and the risk of obesity and diabetes remains controversial [5,6].

Obesity is accompanied by the production of elevated levels of cytokines that induce the development of nonalcoholic fatty liver disease (NAFLD) [7]. Fructose stimulates lipogenesis in a manner independent of insulin, and this, in turn, induces IR through the activation of protein kinase C and the generation of lipid intermediates that favor NAFLD progression [7,8]. Other mechanisms of hepatic IR are the decrease of fatty acid oxidation by inducing mitochondrial dysfunction and the endoplasmic reticulum stress [9]. Decreased fructose consumption may thus be associated with lower hepatic IR in patients with obesity and is believed to be a potential factor in preventing or reversing NAFLD progression.

The current gaps in our knowledge of how fructose intake affects IR make it important to undertake studies under real-life conditions in human populations to increase our understanding of the relationship between these factors. We designed a field trial in patients with overweight and obesity but without diabetes in order to determine whether a low-fructose diet decreases IR independently of a reduction in calorie intake.

## 2. Materials and Methods

The methodological approaches used in this work, the DISFRUTE Study, have been described in detail. What follows is a summary of the most relevant aspects of previously reported methods [10], together with aspects of the methods that were intentionally omitted before to ensure appropriate blinding of the control group.

### 2.1. Objectives

The primary objective of this study was to determine whether decreasing the consumption of foods with high amount of fructose or sucrose led to a decrease in IR after 24 weeks in a population with overweight and obesity, independent of a reduction in calorie intake. The secondary objective was to determine whether, after 48 weeks (24 weeks after the end of the intervention according to the protocol), IR levels remained unchanged compared to the end of the 24-week intervention period.

### 2.2. Study Design

Single-blind field trial (patients, physicians, and nurses in the control group were unaware of which foods had been excluded from the diets in the intervention group) randomized by health care area. The primary outcome measure was HOMA2-IR (Homeostasis model assessment) at baseline, 24 and 48 weeks. The secondary outcomes were body mass index (BMI), waist circumference (WC), waist circumference to height ratio(WC/H) and blood pressure (BP), which were measured at baseline, 4, 8, 12, 16, 20, 24, and 48 weeks, and total cholesterol, high density lipoprotein (HDL) cholesterol, low density lipoprotein (LDL) cholesterol and triglycerides, which were measured at baseline, 24 and 48 weeks.

### 2.3. Sample Size

Because of the lack of consensus regarding the cutoff value of HOMA2-IR that defines the presence of insulin resistance we opted to use bodyweight change. We estimated a mean weight loss of 4 kg (1.43 kg/m^2^ BMI assuming a mean height of 1.67 m for the Canary Islands population), a bilateral significance level of *p* ≤ 0.05, 80% power and a 20% dropout/loss to follow-up rate. A weight loss of 4 kg was estimated based on maximum weight loss in a trial of the effect of lower fructose consumption on bodyweight and IR [11] and because dietary intervention trials aimed at studying weight loss have reported similar amounts of weight loss after 6 months [12]. Accordingly, a minimum of 245 participants per group was the target number.

### 2.4. Participants

Inclusion criteria: Adults, aged between 29 and 66 years, BMI between 29 and 40.99 kg/m^2^. Exclusion criteria: pregnancy, diabetes, any disease, disorder or medications that might affect carbohydrate metabolism.

All subjects gave their informed consent for inclusion before they participated in the study. The study was conducted in accordance with the Declaration of Helsinki, and the protocol was approved by the Ethics Committee of Nuestra Señora de La Candelaria University Hospital, Santa Cruz de Tenerife, Canary Islands-Spain (Reg. number: 160, 23 May 2012). This study was registered with the International Standard Randomized Controlled Trial at www.isrctn.com/ (ID: ISRCTN41579277).

### 2.5. Randomization and Recruitment

Given the impossibility of maintaining a blinding if the participants of the intervention and the control groups were patients from the same health center, to prevent contamination bias a random sampling was applied in which the randomization unit was the health area in the island of Tenerife. To facilitate the recruitment a digital application was designed for physicians or nurses to determine the potential eligibility of participants on the basis of information appearing on the first page of their electronic medical record. Patients were included in the low-fructose diet intervention group (LFD) or the Canary Islands Health Service standard-diet control group (SD) according to the randomized assignment of their health care area.

When individuals agreed to participate in the study (week −2) they were given a concise information pack that included four pages of a food diary and two pages with instructions on how to record foods and beverages consumed during 4 days. At week 0 they provided a blood sample used to measure glucose, total cholesterol, LDL cholesterol, HDL cholesterol, triglycerides, insulin, and Thyroid Stimulating hormone (TSH), and a urine sample to test for microalbuminuria. After blood was obtained, each participant was given a 75-g oral glucose overload, and 2 h later blood glucose, insulin, and lipid profile were determined again. On the day the participants provided the initial blood sample they were interviewed and given a physical examination (to record two blood pressure measurements, weight, height, and waist circumference), and the 4 food diary sheets were reviewed together with the participant.

### 2.6. Intervention

#### 2.6.1. In Both Groups

To make the interventions homogeneous, two months before starting the field work, the research team met with all the collaborating physicians and nurses who would enroll their patients as participants. This working day was divided into two parts. During the first one, the aims and methodology of the study were detailly explained. During the second part, all of them received information about the general aspects of a dietetic prescription. The diets which they were going to use were revised and discussed there. These diets were known by the assistants before the meeting, as they have been recommended by the Canary Health Service for a long time [13]. The general recommendations for a hypocaloric and healthy diet without processed foods were also discussed. It was explained that, once the four diary registers were sent, a member of the research team would analyze it and would send back a summary with specific dietetic recommendations for the patient (with the kcal/day included). The kcal/day in the prescribed diets were calculated as 30 or 40% less than the kcal/day of the participants’ energy requirements for their ideal weight according to age, sex and physical activity [14]. Those included in the SD group were not advised to eliminate sweetened products from the diet (some of which could contain fructose), unless the caloric balance of the diet was affected by them.

#### 2.6.2. In LFD Group

The health professionals from LFD group were especially instructed in how diet fructose was going to be reduced. In this group, the prescribed diet was the same the Canary Health Service recommends, just removing the foods located in the highest fructose quartile. The fruits removed from the diet were: grapefruit, kiwis, apricots, apples, pears, mangos, raisins, dates. The vegetables removed were as follows: hard squash, cabbage, tomatoes, spring onions, zucchini squash, round zucchini, turnips, and leeks. The starchy vegetables removed were sweet potatoes and yams. The type of recommended food was the same in both groups. In the LFD group patients were advised not to eat foods with the highest content of fructose and were encouraged to eat others of the same family (vegetables, fruits, starchy foods, high- protein foods) in order to avoid differences in the calorie intake. A special emphasis was made in removing from the diet sweetened foods, such as soft drinks and other sweetened drinks, sweetened diaries or other sweetened foods labeled as “diet”. LFD group were also explained that some other daily used processed foods also have a fructose and/or sucrose content: roasted coffee, sugary and edulcorated cereals, processed sauce.

#### 2.6.3. General Aspects of Dietetic and Physical Activity Interventions

The intervention began at week 2 and was implemented by the participant’s physician or nurse. All the interventions were individualized and carried out in the doctor or nurse’s office by the same professional all along the study period. An example of a standard and a low fructose diet, as well as of a dietary recommendation, made by a member of the investigation team for a physician or nurse, may be consulted in Appendix A.

Both health personnel and participants in the SD group were unaware of the dietary modifications used in the LFD group. To avoid suspicion or speculation by participants in the SD group that the main focus of the study was sugar intake, fructose and sucrose were not mentioned in the informed consent form.

Leisure time physical activity was recorded as activity during the previous week and the previous 6 months. In this article, we used only the record of the last 6 months because of the excessive number of patients with null physical activity when measuring only one week. If participants reported physical activity equivalent to moderate or brisk walking for 150 min per week or more, they were encouraged to continue this level of activity. If participants reported less activity, they were advised to increase it to the recommended target level. To verify adherence to the diet, physicians and nurses were asked to record 24-h dietary recalls at weeks 8, 12, 16 and 20.

At week 24 a second blood sample was obtained for analysis. Two weeks before week 24 the participants were given a new information pack containing the same materials as the initial pack at the start of the study, and with instructions on how to record foods and beverages consumed during 4 days. On the day of the appointment to provide a new blood sample, the participants met first with the physician or nurse to review their food diaries in detail.

### 2.7. Data Recording and Analysis

A specifically designed case record form (CRF) was used to record personal contact information, socioeconomic class, personal and family antecedents, and to compile data for 24-h recall food consumption and physical activity. For each participant, mean nutritional values were calculated for each food from information in the 4 daily food diaries at baseline and week 24. Nutritional values were estimated mainly from the Mataix Spanish food composition tables for macronutrient, sugar and calorie intakes [15]. The sources for nutritional composition of individual foods or dishes are as Appendix A on line. Fructose and glucose were recorded as total values (free and from sucrose), and for these monosaccharides and sucrose, it was noted whether the source was natural or from added sugar. The Appendix A details which foods or dishes were considered to contain added sugar. All blood tests were done at the Canary Islands University Hospital Complex laboratory with the materials and analytical methods reported in a previous publication [10]. The clinically important minimal difference was established in one measure unit for every variable, and 1% of variation in the energy intake for each nutrient. For blood glucose levels, it was understood as clinically relevant when the participants improved their previous glucose tolerance status.

### 2.8. Statistical Analysis

The results for continuous variables are expressed as the mean ± standard deviation or median and range, and those for categorical variables are expressed as relative frequencies and its 95% confidence intervals (95% CI). Student’s t test for paired samples was used to analyze changes between week 0 and week 24 in the main outcome variables in each group (HOMA2-IR, fasting blood glucose and insulin, and these same measures after an oral glucose overload; fasting LDL cholesterol, HDL cholesterol, and triglycerides, and these same measures after a glucose overload; waist circumference, waist to height ratio, BMI and blood pressure). The same analysis was done for total calorie intake and individual nutrient intakes.

These analyses were also performed for the variables noted above between week 24 and week 48. Comparisons between the LFD and SD group for variables indicative of IR and nutrient intakes were done with Student’s t test for independent samples when the frequency distribution was normal, or with the Mann–Whitney U test or Hodges-Lehman tests when the distribution was not normal. The differences within and between groups are expressed as mean and its 95% CI, except the cases of no normal distribution which are expressed as median and 25–75 percentiles. Multiple linear regression models were also used in these comparisons adjusting for gender, age, physical activity, caloric intake and variables that differed significantly between groups at baseline (fasting glucose and smoking status). Daily mean nutritional density (expressed as % of energy intake) and calorie intake were calculated from the 24-h recall data for weeks 8, 12, 16, and 20, when 2 or more were available. Mean nutritional density and mean calorie intake values for these intermediate weeks were compared to the values for week 0 with the same bivariate analysis used to compare week 24 to week 0 values. As a sensitivity analysis of individuals lost to follow-up (*n* = 199), we used multiple imputations for the quintile differences between week 24 and week 0 of fructose intake, blood glucose, abdominal waist and waist-to-height ratio.

All hypothesis tests were two-sided with a significance level of *p* ≤ 0.05, and all statistical analyses were done with the SPSS 21.0 statistical package of IBM Co^®^ in a NT Professional PC Windows operating system.

## 3. Results

Participant recruitment and flows are shown in Figure 1. The characteristics of the participants before the intervention are summarized in Table 1 and Table 2, which show that blood glucose concentration and the percentage of smokers were lower in the LFD group compared to the SD group.

The prescribed diets contained 1708 ± 335 kcal/day in the LFD group and 1690 ± 353 kcal/day in the SD group (*p* = 0.687). Table 3 shows that from week 0 to week 24, HOMA2-IR, glucose and insulin values and anthropometric measures improved in both groups. The changes in nutritional variables were likewise similar in both groups, except for the greater decrease in fructose intake and MUFA in the LFD group, and the larger increase in protein intake in this group. The decrease in total fructose intake was attributable to the lower consumption of added fructose. While the added fructose and MUFA reductions were observed within and between groups, both in percentage of calorie intake and grams, the increase in percentage of calorie intake for proteins within groups is not observed in grams. Within groups, a reduction of protein intake–in grams–was detected (LFD week 24–week 0: −5.2, 95% CI −8.5, −1.8; SD week 24–week 0: −8.4, 95% CI −12.6, −4.1). Moreover, there were no differences between groups (3.2, 95% CI −2.2, 8.6). The differences between the prescribed kcal/day at the beginning of the intervention and self-reported kcal/day in week 24 were 354 ± 382 in LFD group and 328 ± 378 in SD group (*p* = 0.538). In addition, larger decreases in waist circumference, waist circumference/height ratio and fasting blood glucose were seen in the LFD group.

Table 4 shows the results for comparisons between mean calorie intakes and % of energy intake for each nutrient calculated from the 24-h recalls at intermediate weeks 8, 12, 16 and 20 and mean values from the 4 daily diet diaries at week 0. These data confirmed the results of comparisons between week 24 and week 0 with regard to added fructose and protein intake, but also disclosed a larger decrease in added glucose intake (and consequently in added sugars) along with a larger decrease in calorie intake in group LFD. However, the difference between groups in calorie intake between week 24 and the intermediate weeks was not statistically significant (mean 74.4, 95% CI −18.7, 168.5).

The results for the primary and secondary outcomes for differences between week 48 and week 24 are shown in Table 5. There were 77 participants who reached week 48 in both groups, which represents 63.6% and 65.3% respectively of those who reached week 24. The values of week 0 for these participants are also shown in this table as reference. The decreases in HOMA2-IR and waist circumference seen in both groups at week 24 were maintained only in the LFD group. During this period the number of visits to the doctor or nurse in both groups for any reason was similar but the visits related to the study (any registry of blood pressure, weight, waist circumference or dietary intervention) were more frequent in the LFD group (Percentiles 25, 50, 75 = 2, 3, 5 vs. 1, 2, 4; *p* = 0.043). There was a decrease in diastolic blood pressure between week 24 and 48 in the LFD group, which was not observed in SD group. There were no differences in lipids between groups.

Table 6 summarizes the regression coefficients and confidence intervals obtained in the multivariate analyses for the primary outcome variables and secondary outcome anthropometric variables. These results corroborated the differences in fasting glucose and waist circumference seen in the bivariate analysis.

There were not observed differences between groups in relation to the changes of the glucose tolerance status between week 24 and week 0. As a sensitivity analysis for individuals lost to follow-up (*n* = 199), we used multiple imputations in the quintile differences between week 24 and week 0 for intake of fructose, blood glucose, abdominal waist, and the waist-to-height ratio; this analysis attenuated the differences when missing data were imputed to quintiles 1 and 2, although it remained significant (*p* < 0,05) when quintiles 3, 4, and 5 were imputed.

Only 5 adverse events were observed, all in the LFD group: 3 participants reported constipation, 1 hypotension, and 1 general weakness. All events were transitory, and all 5 participants completed the study. No adverse events were reported in the SD group.

## 4. Discussion

After 24 weeks of intervention, the participants of the LFD group showed a similar decrease in HOMA2-IR to those who received the SD. However, they had a larger decrease in their waist circumference and fasting blood glucose concentration. During the intervention period, there was a greater decrease in fructose consumption and a greater increase in protein consumption in the LFD (−1.4% and +1.4%, respectively, of energy difference). The decrease of total fructose was mainly produced because of the decrease of added fructose. However, the increase in protein consumption was relative, as it was only observed as percentage of calorie intake. This was not observed when the analysis was performed in grams. At week 0 the energy intake from added fructose was higher than energy intake from natural fructose. This turned to the opposite at week 24, therefore the consumption of natural fructose was higher than added fructose in both groups. Between groups, there were no differences in the natural fructose increment. However, at the end of the intervention, there was a greater decrease in the consumption of added fructose in the LFD group, even though the elimination of processed foods was recommended in both groups to avoid differences in the calories consumed. We believe that the previously mentioned decrease in added fructose consumption was an effect of the intervention in the LFD group, since LFD physicians and nurses had more knowledge of industrial foods with higher fructose content. The recommendation of substituting certain natural foods for others of the same type but with less content of fructose could have led these health professionals to make a greater emphasis on the recommendation of abandoning foods with added fructose than the SD group. We think that this is reinforced by the fact that, comparing between groups in the most compliant participants—those who answered to at least two 24-h recalls during the follow up (Table 4)—we could observe, not only a greater reduction of added fructose, but also greater reduction of added glucose, as well as greater calorie intake reduction in the LFD group. Unexpectedly, we found lower consumption of MUFA in the LFD group, but we did not consider it plausible that it contributed to decrease waist circumference or fasting blood glucose, since the decrease in MUFA, with similar decrease in total fat and calories, either worsens [16] or has a neutral effect on IR [17].

The lesser waist circumference observed in the LFD group would reflect a larger decrease in abdominal fat [18]. It has been described that the decrease of fructose consumption reduces visceral fat and hepatic fat in obese children [19]. The mechanism that would explain how fructose consumption increases visceral fat is not clearly known. It has been suggested that when fructose is metabolized in subcutaneous fat, inflammation takes place. Such inflammation would lead to an increase in intracellular cortisol, which function would be to squelch it. Cortisol would also lead to an increased flux of fatty acids from subcutaneous adipocytes, those being a source of fat storage in visceral fat tissue [20]. In our study, the reduction of visceral fat would mean a decrease in free fatty acids release to the portal system, as well as a decrease in proinflammatory adipokines and an increase in adiponectin [21]. Therefore, there would be a reduction in fat storage in the liver, contributing to a less hepatic insulin resistance, which would lead to a decrease in the hepatic de novo lipogenesis and gluconeogenesis. This could explain the larger decrease in fasting glucose in the LFD group due to a reduction of hepatic glucose production [22,23].

It is important to note that an energy decrease of only 1.4% in the LFD group with respect to the SD group in fructose (0.9% of added fructose) has been enough to obtain the metabolic improvements. This suggests that a small decrease of added fructose in a sustained manner may be enough to achieve such improvements.

### 4.1. Protein Consumption

In relation to the higher protein consumption observed in the LFD group, this has been produced only as percentage of total calorie intake, but not as grams of proteins. The effects of proteins on body system and appetite are more related to absolute rather than relative amounts, mainly when a reduction of calorie intake is produced. [24]. Anyway, in our study, Table 6 includes both MUFA and protein in the multivariant analysis. Therefore, it seems unlikely that the protein intake was responsible for the decrease in abdominal waist and blood glucose.

### 4.2. Calorie Intake, Added Glucose and Lipids

Assuming the bias of comparing the mean of 2,3 or 4 recalls of intermediate weeks with four diary registers in week 0, we have observed in this case that the difference of consumption of added glucose and calorie intake between the intermediate weeks and week 0 was higher in the LFD group. These differences were not observed between week 24 and week 0 although a trend was observed. This trend to a lower energy intake in this group could be mediated by the metabolic effect of the fructose reduction itself. Unlike glucose, fructose intake does not acutely stimulate insulin secretion, which would attenuate the stimulation of leptin and probably the inhibition of ghrelin, affecting the regulatory action of these hormones on the energy balance in the central nervous system [25,26,27]. In addition, it has been shown that the consumption of glucose, but not of fructose, reduces the activation of the brain regions that regulate the appetite [28]. Altogether it could explain the tendency to reduce calorie intake in the LFD group. In order to consider the possible influence of both calorie intake and added glucose on our findings, both were introduced in the multivariate analysis between week 24 and 0 and there no were observed changes with respect to the findings of the bivariate analysis. Thus, we think that the calorie intake and added glucose have not had an influence on the observed differences between groups in fasting blood glucose and waist circumference. On the other hand, the absence of a relationship between groups in cholesterol, triglycerides, HDL and LDL cholesterol after the intervention coincides with what has already been described [29].

### 4.3. Changes between Week 24 and Week 48

The absence of variation in HOMA2-IR and waist circumference in the LFD group with respect to the SD group between 24 and 48 weeks was not expected. We think that the greater number of visits to the physician or nurse’s office for reasons related to the study were, probably, the main cause for this finding.

### 4.4. Strengths and Limitations

As a strength, we emphasize that this study has been carried out in primary care patients who live in their environment and intake fructose and other nutrients under real life conditions. Our study main limitation is that, because of the little interest that obese patients have shown to participate and the high dropout rate, we have not reached the proposed sample size. This has decreased the power of our study. However, we have still found significant differences that are clinically important and were corroborated in the sensitivity analysis that included missing individuals. The randomization by geographical area, instead of simple randomization, could be understood as a limitation too. This randomization system was used to avoid contamination bias due to the small geographical zone where the field work was carried out. Keeping doctors, nurses and participants blinded in the SD group would not have been possible. Another limitation may be that we assumed that decreasing the consumption of fructose implies metabolically the opposite to increase its consumption. Almost all studies searched for demonstrating the effect of increased fructose consumption on IR and not the effect of the decrease [4]. Of these studies, those that related more specifically the increase in fructose consumption with hepatic insulin resistance have been performed in healthy patients with normal weight, comparing both isocaloric diets [30,31,32] and high-calorie diets [31,33,34,35], lasted less than 4 weeks and used liquid fructose. There was a difference of about 350 kcal/day between the prescribed and the self-reported diets in week 24 within, not between groups. We think that this may reflect an under-reported of the real intake which has been described in obese people [36]. The evidence at present is insufficient to claim that four diaries are enough to provide a reliable analysis of fructose and sucrose. However, most studies used three or four food diaries as their method of measurement [37]

Our study only measured macronutrients and sugars. The influence of vitamin D [38], Zinc [39] and magnesium [40] as relevant factors in insulin sensitivity has been described. It has also been shown that higher contents of minerals such as potassium, calcium and magnesium could reduce the effects of fructose metabolism [41]. We have not considered trace minerals, or vitamins as possible confusing factors in the design of the study. Considering our source of data of the nutrients, we did not, and would not have been able to determine the values of the previously mentioned micronutrients in processed food. In relation to the participants’ compliance with the diets, assuming the commented bias of comparing 24 hour recalls of the intermediate weeks to daily records in week 0, the fructose consumption difference observed between week 24 and 0 appears to have also occurred during the follow-up.

## 5. Conclusions

We could not demonstrate that the reduction of fructose consumption in the context of a hypocaloric diet in non-diabetic overweight and obese patients decreases HOMA2-IR. However, it produced a decrease in waist circumference and fasting blood glucose. The former is related to a decrease in abdominal and liver fat, which would lead to a decrease in hepatic insulin resistance, which could explain the decrease in fasting blood glucose. In overweight and obese non-diabetic primary care patients a small decrease in the consumption of added fructose in a sustained manner may be enough to achieve metabolic benefits. New studies with larger number of patients are needed to corroborate our results and assess whether the benefit obtained would be limited to hepatic insulin resistance.

## Figures and Tables

**Figure 1 nutrients-12-01149-f001:**
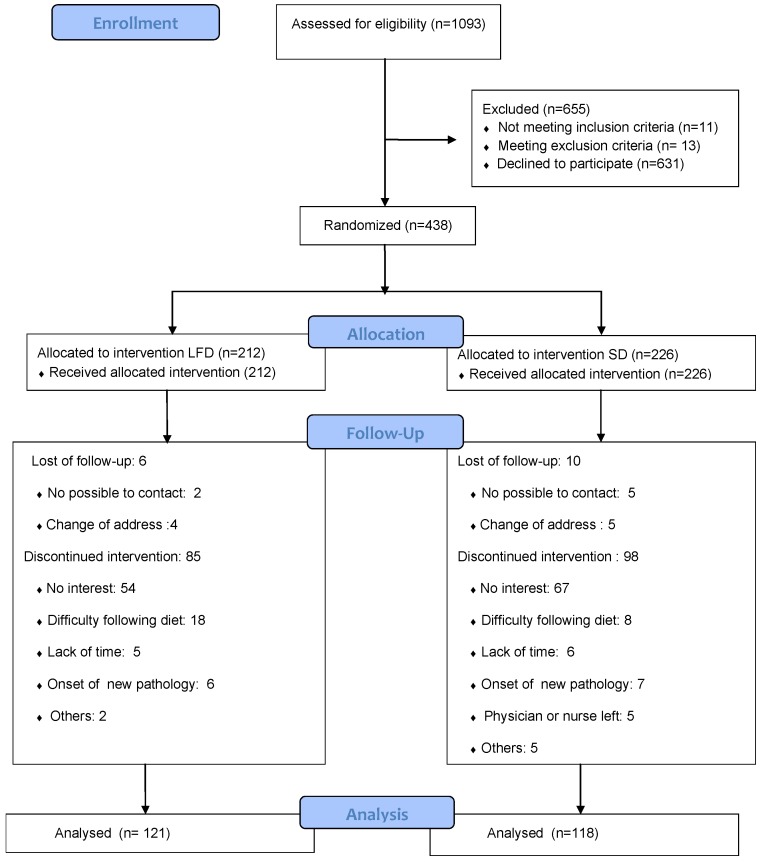
Flow diagram.

**Table 1 nutrients-12-01149-t001:** Baseline Characteristics of the Study Participants.

Characteristic	Began the Study *			Completed Week 24 **		
	All (438)	Low Fructose Diet (212)	Standard Diet (226)	All (239)	Low Fructose Diet (121)	Standard Diet (118)
Age (years)	47.2 ± 8.6	46.3 ± 8.4	48.0 ± 8.7	47.9 ± 8.6	47.5 ± 8.0	48.2 ± 9.1
Women (%)	293 (66.9)	141 (66.5)	152 (67.3)	152 (63.6)	79 (61.9)	73 (65.3)
Height (cm)	164.9 ± 9.5	165.1 ± 9.5	164.8 ± 9.5	164.5 ± 9.9	164.8 ± 10.2	164.3 ± 9.5
Weight (kg)	94.4 ± 13.5	95.0 ± 13.6	93.9 ± 13.4	92.9 ± 13.1	92.8 ± 12.7	93.1 ± 13.6
BMI (kg/m^2^)	34.6 ± 2.9	34.8 ± 3.0	34.4 ± 2.8	34.2 ± 2.8	34.3 ± 2.8	34.2 ± 2.7
Waist circumference (cm)	108.2 ± 9.2	108.6 ± 9.1	107.7 ± 9.3	107.7 ± 8.9	107.9 ± 8.7	107.5 ± 9.1
Waist circumference/height ratio	0.656 ± 0.050	0.659 ± 0.051	0.654 ± 0.050	0.656 ± 0.050	0.657 ± 0.050	0.654 ± 0.050
Hypertension (%)	163 (37.2)	75 (35.4)	88 (38.9)	90 (37.7)	45 (37.2)	45 (38.1)
Hypercholesterolemia (%)	180 (41.1)	95 (42)	85 (40.1)	105 (43.9)	52 (43.0)	53 (44.9)
Familial antecedent of diabetes (%)	222 (51.3)	116(54.7)	106 (48)	136 (56.9)	68 (56.2)	68 (57.6)
Smoking status (%)						
-Current smoker	52 (11.9)	18 (8.5)	34 (15.0)	23 (9.6)	3 (2.5)	20 (16.9)
-Former smoker	93 (21.2)	47 (22.2)	46 (20.4)	50 (20.9)	28 (23.1)	22 (18.6)
-Never smoked	293 (66.9)	147 (69.3)	146 (64.6)	166 (69.5)	90 (74.4)	76 (64.4)
Marital status (%)						
-Married or cohabitating	339(77.4)	166 (78.3)	173 (76.5)	189 (79.1)	96 (79.3)	93 (78.8)
-Divorced/separated	28 (6.4)	15 (7.1)	13 (5.8)	13(5.4)	10 (8,3)	3 (2.5)
-Widowed	13 (3.0)	7 (3.3)	6 (2.6)	9 (3.8)	6 (5)	3 (2.5)
-Single	58 (13.2)	24 (11.3)	34 (15.1)	28 (11.7)	9 (7.4)	19 (16.1)
Social class (%)						
-Low	152 (34.7)	78 (36.8)	74 (32.7)	95 (39.7)	50 (38.8)	45 (42.0)
-Medium	153 (34.9)	78 (36.8)	75 (33.2)	76 (31.8)	37 (31.1)	39 (33.6)
-High	126 (28.8)	54(25.5)	72 (31.9)	64 (26.8)	32 (29.6)	32 (27.6)
-Information not provided	7 (1.6)	2 (0.9)	5 (2.2)	4 (1.7)	2 (1.5)	2 (1.6)
Blood pressure mmHg						
-Systolic	128.3 ± 15.7	129.1 ± 15.2	127.6 ± 16.1	130.2 ± 15.6	130.1 ± 15.9	130.3 ± 15.3
-Diastolic	81.4 ± 9.8	82.4 ± 10.4	80.5 ± 9.1	82.2 ± 9.8	82.6 ± 10.7	81.7 ± 8.6
Physical activity kcal/day						
-Previous 6 months	287.7 (0–6562.2)	312.3 (0–6227.6)	262.6 (0–6562.2)	285.4 (0–6562.2)	310.7 (0–6227.6)	264.4 (0–6562.2)

Quantitative variables shown as the mean ± standard deviation for normal distribution and the median (range) for non-normal distribution. BMI: body mass index. ***** Age, *p*=0.034; diastolic blood pressure, *p* = 0.041. ****** Smoking status, *p* = 0.001; marital status, *p* = 0.034; married or cohabitant vs. all other categories, *p* > 0.05. BMI: body mass index.

**Table 2 nutrients-12-01149-t002:** Baseline Biochemical Variables of the Study Participants.

	Began the Study *			Completed Week 24 **		
	All (438)	Low Fructose Diet (212)	Standard Diet (226)	All (239)	Low Fructose Diet (121)	Standard Diet (118)
Fasting glucose mmol/L	5.08 ± 0.67	4.94 ± 0.64	5.22 ± 0.67	5.11 ± 0.66	4.98 ± 0.64	5.24 ± 0.66
Fasting insulin (µU/mL)	11.7 (2.3–100.6)	11.8 (3.5–100.6)	11.5 (2.3–62)	11.3 (2.3–62.0)	11.5 (3.5–37.4)	11.3 (2.3–62.0)
HOMA-2IR	0.3 (0–2.2)	0.2 (0–2.2)	0.3 (0–1.5)	0.2 (0–1.5)	0.2 (0.1–0.8)	0.2 (0–1.5)
75 g OGTT Glucose mmol/L	6.34 ± 2.08	6.18 ± 2.10	6.49 ± 2.06	6.45 ± 2.30	6.39 ± 2.40	6.58 ± 2.20
75 g OGTT Insulin (µU/mL)	63.2 (6.2–300)	59.5 (8.7–300)	66.2 (6.2–300)	65.5 (7.4–300)	61.3 (13.1–300)	66.9 (7.4–300)
Cholesterol mmol/L						
Total	4.97 ± 0.62	4.98 ± 0.91	4.97 ± 0.94	4.91 ± 0.90	4.90 ± 084	4.91 ± 0.98
LDL	2.99 ± 0.78	3.03 ± 0.80	2.97 ± 0.76	2.94 ± 0.77	2.90 ± 0.76	2.97 ± 0.78
HDL	1.25 ± 0.31	1.24 ± 0.29	1.27 ± 0.32	1.25 ± 0.28	1.28 ± 0.29	1.23 ± 0.27
Triglycerides mmol/L	1.58 ± 0.86	1.56 ± 0.82	1.61 ± 0.89	1.59 ± 0.82	1.58 ± 0.82	1.60 ± 0.83

Quantitative variables shown as the mean ± standard deviation for normal distribution and the median (range) for non-normal distribution. ***** Fasting Glucose, *p* > 0.001 ****** Fasting Glucose, *p* = 0.002. OGTT: oral glucose tolerance test. LDL: low density lipoproteins. HDL: high density lipoproteins.

**Table 3 nutrients-12-01149-t003:** Differences in energy intake, nutrients (% energy), anthropometric variables, blood pressure, biochemical values, and leisure time physical activity between week 24 and week 0 within and between groups.

Variable	Low Fructose Diet (*n* = 121)			Standard Diet (*n* = 118)			Differences between Diets (95% CI)
	Week 0	Week 24	Difference Week 24–Week 0 (95% CI)	Week 0	Week 24	Difference Week 24–Week 0 (95% CI)	
Nutrients and Energy Intake ^a^							
Kcal/day (% of difference and 95% CI)	1900.5 ± 515.3	1354.6 ± 350.6	−545.9 (−28.7 %: −36.7, −20.7)	1841.3 ± 518.2	1362.2 ± 316.4	−479.1 (−26.0%: −33.9, −18.1)	−66.8 (−2.7%: −5.6, 2.0)
Protein	17.3 ± 3.9	22.4 ± 4.8	5.1 (4.3, 5.8)	17.8 ± 3.5	21.4 ± 5.2	3.6 (2.6, 4.6)	1.4 (0.1,2.7) *
Fat	34.5 ± 8.0	29.0 ± 6.4	−5.5 (−7.2, −3.7)	33.9 ± 76.7	29.8 ± 74.5	−4.1 (−6.0, −2.2)	−1.4(−3.9, 1.1)
SFA	9.8 ± 2.4	7.8 ± 2.3	−2.0 (−2.4, −1.5)	9.8 ± 2.9	7.9 ± 2.5	−1.9 (−2.5, −1.3)	−0.1 (−0.8, 0.7)
MUFA	13.2 ± 3.2	11.5 ± 3.3	−1.7 (−2.5, −0.9)	12.7 ± 3.1	12.3 ± 3.8	−0.4 (−1.2, 0.4)	−1.3 (−2.4, −0.2) *
PUFA	4.8 ± 1.7	4.7 ± 1.5	−0.1 (−0.5, 0.3)	4.8 ± 1.8	4.8 ± 1.9	0.0 (−0.4, 0.4)	−0.1 (−0.07, 0.05)
Carbohydrates	49.8 ± 6.8	49.3 ± 7.7	−0.5 (−2.3, 1.2)	49.2 ± 7.5	49.0 ± 8.2	−0.2 (−1.8, 1.5)	−0.3 (−2.7, 2.1)
Starch	25.5 ± 5.8	27.2 ± 7.7	1.7 (0.5, 3.3)	23.9 ± 6.6	24.7 ± 7.7	0.8 (−0.9, 2.5)	0.9 (−1.4, 3.2)
Lactose	3.5 ± 2.0	4.4 ± 2.2	0.9 (0.5, 1.3)	3.9 ± 2.6	4.7 ± 2.3	0.8 (0.4, 1.3)	0.1 (−0.5, 0.7)
Galactose	2.0 ± 1.1	2.6 ± 1.3	0.6 (0.3, 0.9)	2.2 ± 1.3	2.7 ± 1.3	0.5 (0.3, 0.8)	0.1(−0.2, 0.5)
Total sucrose	11.1 ± 4.2	8.2 ± 3.5	−2.9 (−3.8, −2.1)	11.0 ± 4.5	8.9 ± 4.40	−2.1 (−3.1, −1.1)	−0.8 (−2.2, 0.4)
Sucrose in natural foods	2.5 ± 1.6	3.3 ± 1.9	0.7 (0.4, 1.1)	2.8 ± 1.8	3.4 ± 2.1	0.6 (0.1, 1.1)	0.1 (−0.5, 0.7)
Added sucrose	8.6 ± 4.4	5.0 ± 3.5	−3.7 (−4.5, −2.8)	8.2 ± 4.6	5.5 ± 4.1	−2.7 (−3.7, −2.0)	−1.0 (−2.7, 0.3)
Total fructose	10.2 ± 3.3	8.2 ± 3.1	−2.0 (−2.8, −1.3)	10.4 ± 4.1	9.8 ± 3.7	−0.6 (−1.6, 0.3)	−1.4 (−2.6, −0.3) *
Fructose in natural foods	4.5 ± 3.4	5.2 ± 3.0	0.7 (0.0, 1.5)	5.2 ± 4.1	6.5 ± 3.3	1.3 (0.0, 2.2)	−0.6 (−1.8, 0.5)
Added fructose	5.7 ± 2.7	2.9 ± 2.1	−2.8 (−3.3, −2.2)	5.2 ± 2.9	3.3 ± 2.3	−1.9 (−2.5, −1.3)	−0.9 (−1.6, −0.03) *
Total glucose	9.0 ± 2.7	7.4 ± 2.6	−1.6 (−2.2, −0.1)	8.9 ± 3.1	7.8 ± 2.9	−1.1 (−1.7, −0.4)	−0.5 (−1.4, 0.4)
Glucose in natural foods	3.3 ± 2.3	4.1 ± 2.1	0.8 (0.4, 1.3)	3.6 ± 2.4	4.3 ± 2.1	0.7 (0.2, 1.2)	0.1 (−0.6, 0.8)
Added glucose	5.7 ± 2.5	3.3 ± 2.1	−2.4 (−2.9, −1.9)	5.3 ± 2.7	3.6 ± 2.3	−1.7 (−2.3, −1.1)	−0.7 (−0.4, 0.1)
Added sugars	11.4 ± 5.1	6.3 ± 4.0	−5.1 (−6.2, −4.1)	10.5 ± 5.66	6.9 ± 4.5	−3.6 (−5.0, −2.5)	−1.5(−3.0, 0.1)
Fiber#	11.1 ± 6.5	11.7 ± 2.7	0.6 (−0.6, 1.8)	10.6 ± 3.4	13.0 ± 12.4	2.4 (0.1, 4.7)	−1.8 (−4.4, 0.7)
Anthropometric Variables and Blood Pressure							
Weight (kg)	92.8 ± 12.7	86.3 ± 13.5	−6.5 (−7.4, −5.5)	93.1 ± 13.6	87.6 ± 13.3	−5.5(−6.4, −4.6)	−1.0 (−2.5, 0.2)
BMI (kg/m^2^)	34.3 ± 2.8	31.9 ± 3.3	−2.4 (−2.8, −2.0)	34.2 ± 2.7	32.2 ± 3.0	−2.0 (−2.4, −1.7)	−0.4 (−0.9, 0.1)
Waist circumference (cm)	107.9 ± 8.7	100.9 ± 10.3	−7,0 (−8.0, −5.5)	107.5 ± 9.0	102.7 ± 9.3	−4.8 (−5.9, −3.6)	−2.2 (−3.7, −0.7) *
Waist circumference/height ratio	0.66 ± 0.05	0.62 ± 0.06	−0.04 (−0.049, −0.036)	0.65 ± 0.05	0.62 ± 0.05	−0.03 (−0.045, −0.023)	−0.01 (−0.021, −0.005) *
SBP (mmHg)	130.1 ± 15.9	124.5 ± 13.6	−5.6 (−8.4, −2.9)	130.4 ± 15.3	126.2 ± 13.9	−4.2 (−6.5, −1.7)	−1.4 (−5.0, 2.3)
DBP (mmHg)	82.6 ± 10.7	80.4 ± 9.1	−2.2 (−4.0, −0.3)	81.7 ± 8.6	79.2 ± 9.4	−2.5 (−4.2, −0.8)	0.3 (−2.3, 2.7)
Biochemical Values ^b^							
Fasting glucose	4.98 ± 0.64	4,73 ± 0,60	−0.25 (−0.34, −0.17)	5,24 ± 0,66	5,13 ± 0,63	−0.11(−0.21,−0.005)	−0.14 (−0.028, −0.02) *
Fasting insulin	12.5 ± 5.9	11.0 ± 5.8	−1.6 (−2.5, −0.7)	13.3 ± 8.1	11.7 ± 7.4	−1.6 (−2.7, −0.5)	0.0 (−1.4, 1.4)
Fasting HOMA-2IR	0.27 ± 0.13	0.23 ± 0.13	−0.04 (−0.06, −0.02)	0.30 ± 0.19	0.26 ± 0.17	−0.04 (−0.06, −0.01)	0.00 (−0.04, 0.03)
Fasting total cholesterol	4.90 ± 0.84	4.85 ± 0.83	−0.05 (−0.18, 0.08)	4.91 ± 1.03	4.84 ± 0.9	−0.07 (−0.22, 0.07)	0.02 (−0.17, 0.22)
Fasting HDL	1.27 ± 0.29	1.30 ± 0.29	0.03 (−0.01, 0.06)	1.23 ± 0.27	1.26 ± 0.31	0.03 (−0.01, 0.06)	0 (−0.05, 0.05)
Fasting LDL	2.89 ± 0.76	2.90 ± 0.76	0.01 (−0.11, 0.13)	2.97 ± 0.78	2.91 ± 0.73	−0.06 (−0.26, 0.24)	0.07 (−0.09, 0.23)
Fasting Triglycerides	1.58 ± 0.82	1.45 ± 0.84	−0.13 (−0.26, 0.00)	1.60 ± 0.83	1.46 ± 0.79	−0.14 (−0.28, 0.01)	0.01 (−0.2, 0.2)
75 g OGTT glucose	6.39 ± 2.4	5.57 ± 1.53	−0.82 (−1.15, −0.49)	6.56 ± 2.23	5.84 ± 2.03	−0.72 (−1.07, −0.38)	−0.1 (−0.53, 0.43)
75 g OGTT insulin	85.1 ± 71.5	69.2 ± 67.9	−15.9 (−25.1, −6.6))	96.7 ± 82.3	68.8 ± 62.7	−27.9 (−38.6, −17.1)	12.0 (−2.0, 26.1)
75 g OGTT triglycerides	1.54 ± 0.79	1.33 ± 0.68	−0.21 (−0.26, −0.001)	1.54 ± 0.81	1.34 ± 0.68	−0.20 (−0.34, −0.08)	−0.01 (−0.17, 0.16)
Leisure Time Physical Activity ^c^							
Previous 6 months	286 (88, 610)	322 (145, 606)	7.3 (–47, 63)	236 (77, 547)	343 (93, 638)	–11.2 (–25, 97)	9 (–51, 90)

^a^ Nutritional variables: % of daily energy intake as mean± SD (95% CI). ^b^ Biochemical values mmol/L, except insulin µU/mL: mean± SD (95% CI). ^c^ Physical activity (kcal/day): median (percentile25, percentile75). * *p* ≤ 0.05; #Grams per 1000 kcal. SBP: systolic blood pressure. DBP: diastolic blood pressure. BMI: body mass index. HOMA: homeostasis model assessment. SFA: saturated fatty acids. MUFA: monounsaturated fatty acids. PUFA: Polyunsaturated fatty acids. OGTT: oral glucose tolerance test. LDL: low density lipoproteins. HDL: high density lipoproteins.

**Table 4 nutrients-12-01149-t004:** Differences in nutrients (% of energy intake) and mean daily calorie intakes in kcal/day between the intermediate weeks (with at least two 24-h recalls) and week 0.

Caloric Intake and Nutrients	Low Fructose Diet (*n* = 95)			Standard Diet (*n* = 97)			Difference between Diets (95% CI)
	Week 0 (4 Diary Registers)	Weeks 8, 12, 16 and 20 (Mean of 2, 3 or 4 24-h Recalls)	Difference (95% CI)	Week 0 (4 Diary Registers)	Weeks 8, 12, 16 and 20 (Mean of 2, 3 or 4 24-h Recalls)	Difference (95% CI)	
Kcal/day (% of difference and 95% CI)	1904 ± 523	1197 ± 317	−706 (−37.1 %: −46.8, −27.4)	1825 ± 512	1283 ± 328	−542 (−29.7%: −38.8, −20.6)	−164 (−7.4%: −12.6, −2.2) *
Proteins	17.1 ± 3.9	23.5 ± 4.3	6.3 (5.4, 7.1)	17.7 ± 3.4	22.3 ± 4.6	4.6 (3.6, 5.6)	1.7 (0.4, 3.0) *
Fat	34.3 ± 8.1	28.4 ± 6.8	−5.9 (−8.0, −3.6)	34.1 ± 8.2	28.8 ± 6.3	−5.3 (−7.3, −3.3)	−0.6 (−3.6, 2.3)
SFA	9.9 ± 2.4	7.2 ± 2.0	−2.7 (−3.3, 2.0)	9.7 ± 2.3	7.4 ± 2.2	−2.3 (−2.9, −1.5)	−0.4 (−1.4, 0.5)
MUFA	13.2 ± 3.3	11.4 ± 3.3	−1.9 (−2.8, −0.9)	12.9 ± 3.9	11.9 ± 3.7	−1.0 (−1.9, −0.1)	−0.9 (−2.2, 0.4)
PUFA	4.7 ± 1.7	4.9 ± 1.5	0.2 (−0.3, 0.6)	4.9 ± 1.8	4.9 ± 1.7	0.0 (−0.4, 0.5)	0.2 (−0.5, 0.8)
Carbohydrates	50.2 ± 6.7	49.0 ± 7.4	−1.2 (−3.0, 0.7)	49.0 ± 7.8	49.0 ± 7.3	0.0 (−1.9, 1.9)	−1.2 (−3.8, 1.5)
Starch	25.8 ± 5.8	26.2 ± 8.0	0.4 (−1.5, 2.2)	23.9 ± 6.8	24.5 ± 7.6	0.6 (−1.3, 2.6)	−0.2 (−2.9, 2.3)
Lactose	3.5 ± 2.0	4.9 ± 2.3	1.4 (0.8, 1.8)	3.8 ± 2.6	5.2 ± 2.6	1.4 (0.8, 1.9)	0.0 (−0.8, 0.7)
Galactose	2.1 ± 1.1	2.7 ± 1.2	0.6 (0.3, 0.9)	2.2 ± 1.3	2.9 ± 1.4	0.7 (0.4, 1.1)	0.1 ((−0.6, 0.3)
Total sucrose	11.1 ± 4.3	7.7 ± 3.8	−3.4 (−4.5, −2.1)	10.9 ± 4.4	8.2 ± 3.9	−2.7 (−3.7, −1.6)	−0.7 (−2.2, 0.9)
Sucrose in natural foods	2.5 ± 1.6	3.8 ± 1.9	1.3 (0.8, 1.8)	2.8 ± 1.9	3.4 ± 2.0	0.6 (0.1, 1.1)	0.7 (0, 1.3)
Added sucrose	8.5 ± 4.5	3.9 ± 3.5	−4.6 (−5.7, −3.4)	8.1 ± 4.3	48.0 ± 3.6	−3.3 (−4.3, −2.3)	−1.3 (−2.8, 0.2)
Total fructose	10.3 ± 3.4	8.3 ± 3.1	−2.0 (−2.9, −1.0)	10.4 ± 3.5	9.5 ± 3.3	−0.9 (−1.9, 0.1)	−1.1 (−2.4, 0.3)
Fructose in natural foods	4.7 ± 3.6	6.1 ± 3.2	1.4 (0.5, 2.3)	5.4 ± 4.4	6.8 ± 3.4	1.4 (0.5, 2.4)	0.0 (−1.3, 1.3)
Added fructose	5.6 ± 2.6	2.2 ± 2.1	−3.4 (−4.0, −2.7)	5.0 ± 2.8	2.7 ± 2.0	−2.3 (−2.9, −1.7)	−1.1 (−2.0, −0.2) *
Total glucose	9.1 ± 2.8	7.6 ± 2.5	−1.5 (−2.2, −0.7)	8.8 ± 3.2	7.8 ± 2.5	−1.0 (−1.8, −0.3)	−0.5 (−1.5, 0.6)
Glucose in natural foods	3.4 ± 2.4	4.9 ± 2.3	1.5 (0.9, 2.1)	3.7 ± 2.5	4.5 ± 2.2	0.8 (2.2, 1.4)	0.7 (−0.2, 1.5)
Added glucose	5.7 ± 2.5	2.7 ± 2.1	−3.0 (−3.6, −2.3)	5.1 ± 2.6	3.3 ± 2.1	−1.9 (−2.5, −1.3)	−1.1 (− 2, −0.2) *
Added sugars	11.3 ± 0. 5	4.9 ± 4.1	−6.4 (−7.7, −5.1)	10.2 ± 5.3	6.0 ± 3.9	−4.2 (−5.4, −3.0)	−2.2 (−4.0, −0.5) *
Fiber	11.3 ± 7.2	12.9 ± 3.2	1.6 (0, 0.3)	10.8 ± 3.4	12.5 ± 3.4	1.7 (0.9, 2.6)	−0.1 (−0.2, 0.2)

Mean ± SD. (95% CI). * *p* ≤ 0.05. (2 recalls, *n* = 17 in the LFD group, *n* = 6 in the SD group; 3 recalls, *n* = 17 in both groups; 4 recalls, *n* = 61 in the LFD group, *n* = 74 in the SD group). SFA: saturated fatty acids. MUFA: monounsaturated fatty acids. PUFA: Polyunsaturated fatty acids.

**Table 5 nutrients-12-01149-t005:** Differences between week 48 and week 24 within and between groups for primary and secondary outcomes. Values of week 0 as reference.

Outcome	Low Fructose Diet (*n* = 77)				Standard Diet (*n* = 77)				Difference between Diets Week 48–Week 24
	Week 0	Week 24	Week 48	Week 48–Week 24	Week 0	Week 24	Week 48	Week 48–Week 24	
Glucose (mmol/L)	4.98 ± 0.59	4.72 ± 0.51	5.17 ± 0.56	0.45 (0.34, 0.57)	5.19 ± 0.67	5.16 ± 0.68	5.24 ± 0.62	0.08 (−0.07, 0.23)	0.37 (0.18, 0.56)
Insulin (µU/mL)	12.2 ± 5.1	11.2 ± 6.1	10.3 ± 4.8	−0.9 (−2.0, 0.7)	12.4 ± 8.2	11.3 ± 7.9	13.0 ± 10.1	1.7 (0.5, 2.9)	−2.6 (−4.2, −1.1)
HOMA-2IR	0.26 ± 0.10	0.23 ± 0.14	0.22 ± 0.10	−0.01 (−0.03, 0.02)	0.28 ± 0.20	0.25 ± 0.18	0.29 ± 0.231	0.04 (0.01, 0.07)	−0.05 (−0.08, −0.01)
BMI (kg/m^2^)	34.2 ± 2.8	31.9 ± 3.2	32.1 ± 3.1	0.2 (−0.2, 0.6)	34.3 ± 2.8	32.4 ± 3.1	33.0 ± 3.4	0.6 (0.2, 0.9)	−0.4 (−0.9, 1.1)
WC (cm)	107.2 ± 8.4	100.3 ± 10.1	100.3 ± 9.5	0.0 (0.002, 0.01)	108.2 ± 9.5	103.3 ± 9.4	104.5 ± 9.1	1.2 (0.2, 2.1)	−1.2 (−2.5, 0.1)
WC/H ratio	0.655 ± 0.05	0.614 ± 0.056	0.614 ± 0.053	0.0(−0.006, 0.006)	0.658 ± 0.05	0.628 ± 0.053	0.635 ± 0.053	0.007 (0.001, 0.013)	−0.007 (−0.02, 0.001)
Weight (kg)	92.3 ± 12.3	85.8 ± 13.4	86.2 ± 12.8	0.4 (−0.5, 1.3)	93.5 ± 14.1	88.4 ± 13.5	89.9 ± 14.1	1.5 (0.6, 2.5)	−1.1 (−2.5, 0.2),
Total cholesterol (mmol/L)	4.93 ± 0.79	4.86 ± 0.86	5.17 ± 0.88	0.31 (0.16, 0.47)	4.77 ± 0.88	4.78 ± 0.83	5.04 ± 0.94	0.26 (0.1, 0.43)	0.05 (−0.18, 0.27)
HDL (mmol/L)	1.25 ± 0.28	1.28 ± 0.30	1.36 ± 0.34	0.08 (0.03, 0.13)	1.22 ± 0.28	1.25 ± 0.30	1.32 ± 0.33	0.07 (0.02, 0.11)	0.01(−0.06, 0.08)
LDL (mmol/L)	2.92 ± 0.75	2.88 ± 0.79	3.08 ± 0.82	0.20 (0.06, 0.33)	2.87 ± 0.66	2.86 ± 0.58	3.02 ± 0.82	0.16 (0.02, 0.29)	0.04 (−0.15, 0.23)
Triglycerides (mmol/L)	1.66 ± 0.92	1.55 ± 0.94	1.48 ± 0.80	−0.07 (−0.25, 0.11)	1.58 ± 0.84	1.42 ± 0.67	1.48 ± 0.88	0.06 (−0.07, 0.19)	−0.13 (−0.35, 0.38)
SBP (mmHg)	129.8 ± 17.0	124.0 ± 14.2	123.0 ± 14.1	−1.0 (−4.1, 2.1)	128.2 ± 14.6	125.2 ± 14.7	126.3 ± 14.7	1.1 (−1.7, 3.9)	−2.1 (−6.2, 2.1)
DBP (mmHg)	82.3 ± 11.6	80.3 ± 9.9	77.8 ± 9.5	−2.5 (−4.3, −0.7)	80.7 ± 8.5	79.1 ± 9.3	81.2 ± 13.4	2.1 (−0.8, 5.0)	−4.6 (−8.0, −1.2)

Mean ± SD for weeks 0, 24 and 48. Mean (95% CI) for difference at week 24 to 48 within and between groups. WC: waist circumference. H: height. BMI: body mass index. HOMA: homeostasis model assessment. LDL: low density lipoproteins. HDL: high density lipoproteins. SBP: systolic blood pressure. DBP: diastolic blood pressure.

**Table 6 nutrients-12-01149-t006:** Multiple linear regression models: differences between week 24 and week 0 in primary and secondary outcomes as dependent and group as explanatory variable.

Outcomes ^a^	Group (SD = 0, LFD = 1).	
Primary	B (95% CI)	*p*
HOMA2-IR	−0.004 (−0.040, 0.032)	0.822
Fasting glucose (mmol/L)	−0.27 (−0.39, −0.14)	<0.001
Fasting insulin (µU/mL)	−0.03 (−1.6, 1.5)	0.971
75 g OGTT glucose (mmol/L)	−0.22 (−0.72, 0.28)	0.382
75 g OGTT insulin (µU/mL)	14.9 (−0.3, 30.1)	0.054
Secondary	B (95% CI)	*p*
Weight (kg)	−0.1 (−1.6, 1.3)	0.665
BMI (kg/m^2^)	−0.08 (−0.61, 0.45)	0.772
WC (cm)	−1.7 (−3.3, −0.062)	0.043
WC/H ratio	−0.01 (−0.020, −0.001)	0.035

^a^ Adjusted for age, sex, smoking status, fasting glucose at week 0, difference in energy intake between week 24 and week 0, difference in physical activity during the 6 previous months, difference in protein intake between week 24 and week 0, difference in MUFA between week 24 and week 0, and difference in added glucose between week 24 and 0.

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
