# Peer review of "Decreased Consumption of Added Fructose Reduces Waist Circumference and Blood Glucose Concentration in Patients with Overweight and Obesity. The DISFRUTE Study: A Randomised Trial in Primary Care"

_nutrients, 2020, doi:10.3390/nu12041149_

Round 1

Reviewer 1 Report

The authors present an ambitious field trial aimed at determining whether dietary fructose reduction in non-diabetic individuals with overweight and obesity would reduce insulin resistance as measured by HOMA2-IR.  Although they were unable to achieve their targeted trial completion criteria (n=245 per group) which limited the power of their study, they reported observing significantly reduced fasting glucose and waist circumference after 24 weeks (22 weeks of intervention), associated with a self-reported reduction of added fructose consumption assessed by diet diary and recall methods.

I have several comments that may improve the presentation and comprehension of the manuscript. 

  1. In lines 82-84 the authors state, …“The secondary objective was to determine whether after 48 weeks (24 weeks after the end of the intervention according to the protocol), IR levels remained unchanged compared to the end of the 24-week intervention period.” Elsewhere in the manuscript (e.g., title to Table 6), the authors indicate that anthropometric measures of weight, BMI, WC and WC/HT ratio are the secondary outcomes. Are objectives and outcomes the same as it would seem using other examples? E.g., in lines 80-82, the authors state the primary objectives…and later in the manuscript they state these same objectives as outcomes…so, that is consistent.  This seems a minor point, but it would help to be consistent throughout the manuscript in identifying secondary objectives/outcomes.
  2. On line 95 the authors state that they expected a 4 kg weight loss when estimating sample size. What was this based on and how was this calculated? Was this estimate related to the 30-40% calorie reduction recommendation that was the intervention target? 
  3. In lines 139-140, the authors state, “The kcal/day in the prescribed diets were calculated as 30 or 40% less than the kcal/day of the participants’ energy requirements for their ideal weight according to age, sex and physical activity.” What equation was used to estimate usual intake?  When they say 30-40% less, why is there such a large range?  Were the recommendations given on an individual basis or by treatment group? This should be clarified.
  4. Line 147. The authors state that participants in the LF group were advised to just replace foods in the highest fructose quartile. While the authors listed the foods that participants were recommended to eliminate, they did not specify examples of what foods should replace those that were eliminated in order to maintain calories equivalent to the SD group.  It would help to be more specific about what instructions were given to participants in this regard.
  5. Lines 167-168. The authors explain that the dietary recalls were compared to week zero diaries as a way to estimate dietary adherence.  Why did they not compare the recalls to the diaries at the end of 24 weeks as well? 
  6. Table 2 format. It would be easier for the reader to compare the intervention effect if the columns were re-organized so that the week zero values and the week 24 values were side by side for each diet, vs. organizing them with LF next to SD.  Since most of the changes occurred over time of intervention, this would highlight the effect of the intervention.
  7. Lines 231-232. The authors present the daily calorie prescriptions for the two treatment groups.  What do these number mean in relation to the earlier statement that the diets provided 30-40% less than daily energy needs?  What do they mean in relation to the diet diary and recall numbers reported?  Should these prescribed values be compared to the self-reported values as an estimate of adherence?  If these intakes were actually achieved during the study what would have been the expected weight loss (e.g. using Kevin Hall’s model estimates). 
  8. In Table 3, why do the authors include values for cholesterol, HDL, and LDL following a 75 gram oral glucose load? Would these variables be expected to change acutely?  If not, these could be eliminated to shorten the table.  Also, the values for physical activity after 24 weeks vs. week 0 are not statistically different, but the mean values were much lower after 24 weeks.  Seems odd, especially given the authors reported that the participants were mostly very sedentary.  Any explanation?
  9. Line 248. The authors note that the comparison of recalls to week zero data indicate a significantly larger drop in energy intake from week 0 in the LF group compared to the SD group.  In Table 3, comparing diary data between weeks 0 and 24, there was no difference between the groups.  Although later, the authors show that the multivariate analysis ruled out that any difference in calorie intake did not explain the observed changes in WC and fasting glucose, the authors should say something about the apparent conundrum re. calorie intakes and their primary objective of looking at fructose reduction in the absence of a change in calorie intake.  Either the recall data vs. week 0 is to be their main comparison for calorie intake, or it should be eliminated for that purpose and the week 24 diary data only, should be used.
  10. Table 5. For triglycerides, the 95% confidence intervals are missing for the LF diet, wk 48-wk 24 value.
  11. Line 283. The authors reference the increase in protein consumption in the LF group relative to the SD group.  Did protein intake actually increase on a grams/day basis vs. just a %kcal basis?  This is important as the effects of protein on body systems and appetite may be related to absolute amount vs. relative amount.  It would help to add the absolute protein intakes in this context.
  12. Line 295. The reference to recommending certain substitution foods with less fructose content reinforces the point made in #4 above…that it would be good to include examples in the methods of such replacement foods so the reader better understands the real world practicality of adopting and adhering to the LF diet that was recommended.
  13. Line 316. The authors use the terminology “flux of fatty acids”.  Flux from where?  The liver?  Peripheral adipose tissue lipolysis?
  14. Lines 341-343. The authors suggest that the “increase” in protein intake may have caused the reduced calorie intake in the LF group.  See point 11 above and consider reworking this section given that protein intake may not have been different on an absolute basis.  This section also re-raises the potential logic problem mentioned in point 10 above.  Was calorie intake on the LF diet reduced or not, compared to the SD?
  15. Line 358, section on limitations. The authors should at least acknowledge the inherent limitations of using diet records and recalls for examining nutrient intake.  While there is still vigorous debate about this among nutrition scientists and epidemiologists, it is worth keeping in mind that the quantitative estimates presented in the table throughout this manuscript should not be taken at face value, and the estimates of what changes in dietary intake are necessary to produce an observed effect should be used cautiously.  

Reviewer 2 Report

Domínguez-Coello and collaborators present an ambitious study aiming at determining in a relatively large population – and in a real-life setting -  the metabolic and physiological consequences of dietary reduction of fructose.

The study is commendable, inasmuch it aims at determining the potential beneficial effects of fructose reduction in the diet. It is, indeed, demonstrated that fructose in a stronger inducer of insulin resistance as compared to glucose.

While nutritional studies assessing the negative effects of fructose administration are numerous, and indeed support the metabolically detrimental role of fructose, attempts to reduce fructose intake in nutritional trials are scarce, and this study would therefore provide a much needed novel information.

I have, however, very major concerns about the design and execution of the study.

  • The trial is merging two interventional situations, namely, a substantial decrease in calorie intake for both study groups (LFD and SD) AND decreased fructose intake.
  • As apparent from the data in the tables (in particular table 3), it seems that the reduction in fructose intake in the LFD group is, at best, minimal, and at any instance comparable to the simultaneous reduction in glucose.

Therefore, several problems emerge here:

  • It is unclear whether the differences observed should be imputed to the strict caloric restriction regimen, which likely has a stronger impact than the relatively minor modulations of fructose (but also glucose) intake
  • Besides the effects related to caloric restriction , which is indeed assigned to both groups, fructose and glucose reductions are comparable, making it impossible to assess the specific contribution of fructose reduction.

In addition, nutrients provision is described in a rather unclear way. In table 2, data about fructose are reported as 1) total fructose , 2) free fructose, 3) Total fructose in natural foods, 4) Added total fructose, 5) Free fructose in natural foods, 6) Added free fructose (and the same applies to glucose). It is unclear for the reader which value would bring the most of understanding, and one would argue on the precision for the evaluation of parameters such as  “Total fructose in natural foods” which likely comes from nutritional questionnaires with their intrinsic limitations.

Table 4 is also difficult to apprehend: in the central column for both standard diet and LFD there is a one single value that reports observations in intermediate weeks (week 8,12, 16, 20). One would expect 4 values.

Overall, the authors certainly did a very ambitious study, but the overall feeling is that they tend to overemphasize the observations relative to fructose consumption, and the tables are presented in an overwhelming way, with far too much information.

It seems to me that the key advance in this study is the strong effort that has been made in submitting the study population to caloric restriction, and it is likely that the most of the metabolic parameters that become ameliorated are a consequence of the caloric restriction intervention.

The impact if fructose administration is not fully supported by the data, and in this respect the authors should limit the emphasis on this aspect in their investigation and in the manuscript.

Reviewer 3 Report

 When you use the abbreviation for the first time, write the full name. ex) HDL, LDL,TSH

  And I recommend the abbreviation when use the common medical words. ex) WC, WHtR, BP 

Please provide a reference. line 61, 315

 Please move the titel of table 2 above that table.

 I think there was significant difference in the total energy intake between two group according to table 4 (during the study). 

How can you explain low fructose diet decrease hepatic insulin resistance and reduce WC, FBG.

Round 2

Reviewer 1 Report

The authors have satisfactorily addressed the reviewer's comments.  There are a few remaining minor points that could be addressed, shown below.

line 143:  Add the reference from the Spanish Ministry of Health Guideline, 2008, as the source for recommended calorie intakes by age and sex.  

line 264:  Consider adding a sentence stating that the calorie intake values reported at 24 weeks via diaries were not different from the intermediate values collected by recall.

lines 366-367:  The authors state that the consumption of added glucose and calories was higher in the LFD group at intermediate weeks vs. time 0.  This is not consistent with the data in table 4, which shows these values to be lower at the intermediate time points than at time zero.  Perhaps the authors meant to say that the difference between week 0 and the intermediate time points for these variables was greater in the LFD group compared to the SD group?  Otherwise, the statement on line 368 about a trend for decreased energy intake would not make sense.  

line 379:  add the word "with" before the word "respect"

line 382: capitalize hdl and ldl to make consistent with the rest of the manuscript  

Author Response

Comments and Suggestions for Authors

The authors have satisfactorily addressed the reviewer's comments. There are a few remaining minor points that could be addressed, shown below.

POINT 1: line 143: Add the reference from the Spanish Ministry of Health Guideline, 2008, as the source for recommended calorie intakes by age and sex.

RESPONSE 1: We have added that reference.

POINT 2: line 264: Consider adding a sentence stating that the calorie intake values reported at 24 weeks via diaries were not different from the intermediate values collected by recall.

RESPONSE 2: We have added “However, the difference in calorie intake between week 24 and the intermediate weeks between groups was not statistically significant (mean 74.4, 95%CI −18.7, 168.5).”  We have added it at the end of the paragraph instead of the line 264 because it seems more appropriated.

POINT 3: lines 366-367: The authors state that the consumption of added glucose and calories was higher in the LFD group at intermediate weeks vs. time 0. This is not consistent with the data in table 4, which shows these values to be lower at the intermediate time points than at time zero. Perhaps the authors meant to say that the difference between week 0 and the intermediate time points for these variables was greater in the LFD group compared to the SD group? Otherwise, the statement on line 368 about a trend for decreased energy intake would not make sense.

RESPONSE 3: Certainly. We have corrected it.

POINT 4: line 379: add the word "with" before the word "respect"

RESPONSE 4: Thanks. It has been done.

POINT 5: line 382: capitalize hdl and ldl to make consistent with the rest of the manuscript

REPONSE 5: Thanks. It has been done.

Please, note that in line 379 we had stated:  “In order to consider the possible influence of both calorie intake and added fructose on our findings, both were introduced in the multivariate analysis…..” We have now changed added fructose to added glucose. It was a mistake.

We have also deleted in the footnotes of Tables 3 and 4 “Added free fructose (no normal distribution): median [Pc25,Pc75]” because it does not sense since we had deleted free fructose, free glucose and free galactose in the body of the Tables.

We are sorry for all that.

Many thank for all your comments and observations. They have improved this manuscript.

Reviewer 2 Report

I have no further comments to present to the authors.

Author Response

Thanks for your review. 

This manuscript is a resubmission of an earlier submission. The following is a list of the peer review reports and author responses from that submission.

Round 1

Reviewer 1 Report

This is a quite interesting manuscript on the consequences of decreased fructose ingestion upon metabolic and anthropometric parameters.

A more detailed description of the results would improve the manuscript.

It is important to consider/calculate the differences of minerals ingestion (both from food and beverages (including water)) among the 2 groups (LFD vs SD) as they can modulate fructose negative impact upon metabolic homeostasis. The other way around, and in line, minerals deficiency by itself can contribute to insulin resistance.

Then these results should be taken into consideration in the discussion.

In a quite interesting article, Pereira et al showed that fructose negative impact upon triglycerides, insulin and leptin levels as well as upon insulin sensitivity was impeded/reduced by mineral ingestion (from mineral rich water).

Examples:

Pereira CD, Severo M, Araújo JR, Guimarães JT, Pestana D, Santos A, Ferreira R, Ascensão A, Magalhães J, Azevedo I, Monteiro R, Martins MJ. Relevance of a Hypersaline Sodium-Rich Naturally Sparkling Mineral Water to the Protection against Metabolic Syndrome Induction in Fructose-Fed Sprague-Dawley Rats: A Biochemical, Metabolic, and Redox Approach. Int J Endocrinol. 2014;2014:384583.

Cruz KJC, de Oliveira ARS, Morais JBS, Severo JS, Mendes PMV, de Sousa Melo SR, de Sousa GS, Marreiro DDN. Zinc and Insulin Resistance: Biochemical and Molecular Aspects. Biol Trace Elem Res. 2018 Dec;186(2):407-412.

Kostov K. Effects of Magnesium Deficiency on Mechanisms of Insulin Resistance in Type 2 Diabetes: Focusing on the Processes of Insulin Secretion and Signaling. Int J Mol Sci. 2019 Mar 18;20(6).

Also, the discussion should be further improved by including the fact that lower waist circumference would mean less adipose tissue draining into the liver molecules that contribute to liver insulin resistance.

Reviewer 2 Report

According to the sample size analysis, a minimum of 245 subjects per group is needed, but, the authors start the study by 212 for the LFD and 226 for the SD. Please explain. This is regardless of the dropout. I am suggesting to divide Table 1 into two tables. First for general questions and second for the biochemical variables. Under 2.6. Intervention section, the authors wrote that the subjects were advised to consume 1000, 1250, 1500, 1750, 2000, 2250, 2500 or 2750 kcals. But please explain how these kcals were assigned to each individual. In addition, if the subjects were instructed to 1000, 1250, 1500, 1750, 2000, 2250, 2500,  2750 kcal, how come the final nutrition intake was only around 1355 kcal? Could it be, that all subjects were advised to consume less than 1500 kcal? Considering the body weight, isn't the dietary(kcal) modification unrealistic? This could be one of the reasons for such a high dropout. Considering the baseline nutrient intake, the overall profile seems to be a pretty normal nutritional intake. 1900kcal of total calorie, with ~35% fat, 17% protein, 48% carbohydrate with less than 10g of fructose, 11g of sucrose, 9 g of glucose (less than 50g of sugar, which is the general guideline of WHO). What is the normal intake of fructose of the country? Do the authors really think that this dietary pattern needs to be corrected? Regardless of much anticipation, the IR was not significantly changed after 24 weeks. What would the authors suggest for these results? The authors elaborated a lot regarding the higher protein intake at the discussion section, but, it is not clear what the authors want to explain. First, the protein intake in the LFD is only 22% which is within a fairly normal range, and second, it is the ratio that was increased, but the amount of protein consumed in gram is not increased much. Also I think trying to explain the result by increase of protein intake is a little bit too risky.

Round 2

Reviewer 1 Report

The article still needs to be improved.

The previous suggestions/comments were not totally addressed.

A more detailed description of the results would improve the manuscript. Please consider the following. Table 1 legend is incomplete. What is missing? BMI.

Table 1 legend: please check the p value shown for married or cohabitant vs. all other categories. Is p=0.53 correct? In case it is correct, there is no need to disclose the value (as it is not significant).

Table 2 legend is incomplete. What is missing? HOMA-2IR, OGTT, LDL and HDL. “The changes in nutritional variables were likewise similar in both groups, except for the greater decrease in fructose intake and MUFA in the LFD group, and the larger increase in protein intake in this group. The decrease in fructose intake was attributable to the lower consumption of added fructose.”

Include/discriminate in what form fructose intake decreased – total fructose, added total fructose and added free fructose.

Mention that the variation pattern for free fructose seemed opposite in LFD vs SD.

“In addition, larger decreases in waist circumference and fasting blood glucose were seen in the LFD group.”

Include not only waist circumference but also waist circumference/height ratio.

Table 3 legend is quite incomplete. What is missing? SFA, MUFA, PUFA, BMI, HOMA-2IR, HDL, LDL and OGTT. Regarding Table 4, the statistical difference seen for added sugars is not mentioned in the body of manuscript (within the results section) neither is the opposite variation pattern seen for starch. Please mention/comment/describe the other variables in Table 5, other than HOMA-2IR and waist circumference. “These results corroborated the differences fasting glucose and waist circumference seen in the bivariate analysis.”

The above sentence needs grammatical correction.

Check the legend of Tables 4, 5 and 6. Regarding the discussion. It should be mentioned/stated/discussed that minerals modulate fructose impact upon metabolism homeostasis.

As the LFD group had dietary restrictions in foods rich in minerals (fruits and vegetables) there could be misinterpretation of the positive results found in the manuscript (hypothetically, the results could have been better for the LFD group if mineral ingestion was kept constant/unaltered). The authors should discuss this. And they should reference this in the discussion, either using or not the article previously suggested (Pereira et al 2014) where is clearly seen that minerals modulate fructose impact.

In the study under review a putative decrease of minerals ingestion by the LFD group could have hindered/reduced/blocked the positive effects of reducing fructose intake.

Besides Zinc many other minerals modulate insulin resistance. Magnesium has also a crucial/prime role in insulin resistance and inflammation (that is now discussed in the manuscript). The authors may choose whatever articles they reference in the manuscript but magnesium must be considered in the discussion. it is not expected that changes in vitamin D have occurred between the two groups, as total fat ingestion does not differ between LFD and SD.

Reviewer 2 Report

The points were very well answered by the authors. 

Congratulations! 

Author Response

Thanks for your review.